# Well-Being Workshops in Eating Disorder Wards and Their Perceived Benefits to Patients and the Multi-Disciplinary Team: A Pilot Study

**DOI:** 10.3390/brainsci9100247

**Published:** 2019-09-23

**Authors:** Katherine Smith, Yasemin Dandil, Claire Baillie, Kate Tchanturia

**Affiliations:** 1Department of Psychological Medicine, Institute of Psychiatry, Psychology and Neuroscience, Psychological Medicine, King’s College London, London SE5 8AF, UK; katherine.a.smith@kcl.ac.uk; 2South London and Maudsley NHS Foundation, London BR3 3BX, UK; yasemin.dandil@slam.nhs.uk (Y.D.); claire.baillie@slam.nhs.uk (C.B.); 3Department of Psychology, Illia State University, Tbilisi 0162, Georgia

**Keywords:** well-being, anorexia nervosa, eating disorders, recovery model, burnout

## Abstract

A more holistic definition of patients’ recovery from eating disorders (EDs) highlights that well-being interventions linked to self-compassion are under-researched and under-utilised. Staff burnout is also common in ED units (EDUs), linked to difficult relationships with patients and poor self-care, and is not well addressed. Therefore we piloted a series of joint well-being workshops to target these issues. Joint workshops were offered to patients (*n* = 55) and the multi-disciplinary team MDT (*n* = 34) in adult ED wards over two years. Experiences were evaluated quantitively and qualitatively. Mood post-workshops increased significantly for both groups (patients: *p* < 0.001, r = 0.49; MDT: *z* = 3.043, *p* = 0.002, r = 0.41), with the feeling that they deserved to take time for self-care (patients: *z* = 2.419, *p* = 0.016, r = 0.31); MDT: *z* = 2.814, *p* = 0.005, r = 0.38). Workshops were found to be enjoyable and highly relevant to well-being, but less useful by patients. Thematic analysis identified six themes: Enjoyment, recovery and well-being, relationships, content, structure and future ideas. Both groups experienced improved mood and increased enjoyment and awareness of well-being. Patient isolation was addressed, and the staff experienced stress reduction and increased productivity. Both groups experienced improved relationships.

## 1. Introduction

Recent years have seen an increased interest in a broader, more holistic definition of recovery in eating disorders (EDs) [1,2].This definition highlights well-being, in addition to the remission of ED symptoms, as a fundamental criterion for ED recovery [1]. Anorexia Nervosa (AN) has a high mortality rate and more than 50% of patients diagnosed still meet the diagnostic criteria 20 years later [3]. In seeking to improve treatments, we need to evaluate the contribution of all options. Recent research suggests that well-being and ED recovery are strongly related [4]. Yet a systematic review showed that using complementary and alternative medicine for well-being, such as yoga and mindfulness, in ED recovery is poorly understood [5]. Well-being is linked to self-care and developing self-compassion [6]. Most patients with an ED have low self-compassion, compounded by a fear of self-compassion or of treating themselves with kindness [7]. Increasing self-compassion relates to lower levels of anxiety, depression, worrying thoughts and fear of failing, and higher levels of happiness, curiosity and social connections [8].

In-patients with an ED have been found to have decreased social connections and increased levels of isolation [9]. Poor social inclusion has been linked to delayed recovery in AN [10]. Within ED treatment programs, the benefits of social interactions have been demonstrated and recognised by patients attending group treatment interventions [11]. Day-to-day social interactions are often strained, especially involving staff and patients at mealtimes, and these interactions have a direct impact on the wellbeing of staff as well as patients [12]. 

Poor well-being in mental healthcare staff is closely related to burnout [13] and instances of this are thought to be significantly higher in EDUs [14]. High rates of relapse and low success rate in ED treatment are thought to be a major reason for burnout and stress in health care professionals [15]. Until recently, burnout was narrowly defined as a medical diagnosis, limiting how it is approached and managed at an organisational level [16]. Since precipitating factors are often beyond the individual’s control, it is more appropriate to reframe burnout as an occupational problem [17], allowing us to ameliorate and treat it [16]. Burnout is a serious problem, and research suggests initiatives to address this have had minimal impact [18]. However, well-being programs in hospitals have been shown to positively impacted clinicians stress and well-being levels a year later [19]. Furthermore, research shows collaborative programmes of joint activities such as drumming between patients and carers successfully enhance the well-being of both [20]. 

This pilot study investigated the effectiveness of joint well-being workshops focused on the aspects of well-being for both patients and the multi-disciplinary team (MDT) in ED units (EDUs). Workshops had multiple well-being aims. Firstly, to provide patients with new skills, insight and awareness to continue their recovery post EDUs, for example, self-compassion and positive communication [21]. Secondly, supporting the MDT and providing stress-related well-being benefits. Thirdly, through the unique feature of bringing patients and the clinical team together, we hoped to facilitate enjoyable and meaningful interactions away from the ward, to shift dynamics, improve relationships and communication, and reduce the tension of mealtimes. 

## 2. Materials and Methods

### 2.1. Design 

To gain meaningful data, the study design was mixed methods, using both qualitative and quantitative approaches. This was of particular importance in studying an illness with an ego-syntonic nature such as AN, where it is very hard as patients often struggle to engage in treatment [22].

### 2.2. Participants 

Between 2017 and 2019, patients and the MDT in adult EDUs at Bethlem Royal Hosptial were offered the opportunity to attend well-being workshops. All upcoming workshops were advertised on the ward with posters and verbally to both patients and the MDT. The MDT were further reminded by e-mail. Participants were adult patients (between the ages of 18–63) with a primary diagnosis of AN, as diagnosed by a consultant psychiatrist based on the Diagnostic and Statistical Manual of Mental Disorders DSM IV [23] and V [24] criteria. All patient participants were receiving treatment in a specialist ED treatment program (ward or residential care), incorporating various psychological, medical and nutritional interventions. Fifty-five patients and 34 members on the MDT on the ward attended, consented and participated in the workshop study. There was no exclusion criteria for the MDT. No information was collected regarding the relationships between the participants. 

### 2.3. Procedures and Measures 

Upon admission to the wards, patients completed audit measures with demographic information, such as the Autism Spectrum Quotent Score, short version (AQ10). Shortly after the introduction of the well-being workshops in 2017, self-report well-being questionnaires and feedback forms were collected from patients and the MDT after all workshops (see Appendix A). Post-pilot, a patient discussion focus group and semi-structured interviews with staff provided further outcome measures.

The well-being questionnaire issued pre (T1) and post (T2) workshops, consisted of five statements relating to taking time for themselves and to mood. Participants were asked to self-rate on a five-point Likert scale where one was “completely disagree” and five “completely agree”. The post-workshop feedback form asked participants to rate aspects of the workshop, with answers on a five-point Likert scales, where five was very good or useful. 

Qualitative data was obtained from various sources (see Appendix A): A feedback form contained open-ended questions about the workshop experience; a fifty-minute focus group with eight patient participants discussed the impact and benefits of the particular wellbeing interventions; and nine MDT attendees took part in individual semi-structured interviews.

### 2.4. Intervention: Well-being Workshops

Due to the pilot nature of this study, these workshops were selected to explore and evaluate satisfaction, relevance and importance to inform future directions for the workshops. Due to limited funding, workshop themes were also dependent on professional facilitators volunteering their time and resources. This also impacted the frequency of the workshops. We wanted external facilitators to reduce bias in relationships. Due to the varied age range of patients, the concept of well-being may vary. We chose themes we believed to be relevant across all age groups. Themes were chosen in advance and patricipants were encouraged to attend. 

We ran a total of eight workshops in teaching environments across the hospital grounds (e.g., the Museum of the Mind) such as

Positive Communication Workshop: Focused on social skills, combining psycho-education materials and experiential exercises. Skills learned included "recognising positive social signals", "expressing positive signals" and "regulating negative emotions". This workshop aimed to help participants positively communicate needs, emphasising the importance of body language, facial expression and voice. 

Colour Workshop: This workshop emphasised the importance of colour and how it can enhance mood and its importance in social situations. 

Make-up Workshop: A professional make-up artist taught attendees techniques through demonstrations. Looks were created for formal day settings (e.g., job interviews) and leisure. 

Yoga Workshop: Yoga focused on breathing and stretching. The instructor was in recovery from an ED and shared her recovery journey story. 

Voice Coaching Workshop: Body and vocal stretches were the focus. Participants also shared happy memories with their partners in the group.

### 2.5. Analysis

Quantitative data was analysed using IBM SPSS Statistics (Version 25.0, IBM Corp., Armonk, NY, USA) for Macintosh, version 25.0. Wilcoxon Signed–Ranks tests were conducted to explore pre and post-workshop differences and effect sizes were calculated [25]. 

Qualitative thematic analysis [26] was conducted on transcribed voice recordings of the patient focus group and MDT interviews, and also on answers provided to the open-ended questions on the feedback forms. Key themes were identified.

### 2.6. Ethical Considerations

Participation by patients and the MDT was voluntary. Informed, written consent was obtained from all participants. Ethical approval was obtained as a service improvement by the National Health Service (NHS) governing body.

## 3. Results

### 3.1. Demographics

Fifty-five patients participated in the study, with a mean age of 29.0 years (SD = 10.7). The mean BMI at the start of treatment was 14.1 (SD = 1.4) and the average duration of the illness was 7.7 years (SD = 8.2). Forty patient participants (81.6%) had a diagnosis of AN restrictive subtype, and nine (18.4%) had a diagnosis of AN Binge–Purge subtype. Fourteen (32.6%) of these had a previous diagnosis of ASD or scored highly (+6) on the AQ10 at admission. Of the total fifty-five patient attendees, twenty attended positive communication and emotion skills workshops, seventeen patients attended make-up workshops, five attended the voice coaching and thirteen attended the yoga workshop.

In total, thirty-four MDT took part in the well-being workshops. Fourteen (41.2%) were therapists, thirteen (38.2%) were nurses, three (8.8%) Occupational Therapists and four (11.8%) were administrators. Eleven attended make-up workshops, ten attended positive communication and emotion skills workshops, ten attended voice coaching and three attended the yoga workshop.

### 3.2. Quantitative Results

#### 3.2.1. Workshops

Quantitative data was collected from six workshops. This was made up of one run of the positive communication workshop, one run of the voice coaching workshop, and two runs of both the yoga workshop and the colour workshop, all in different months. 

#### 3.2.2. Patients

Fifty-three (96%) patients provided data for quantitative analysis, in the form of T1/T2 well-being questionnaires (*n* = 30, 55%), or feedback forms (*n* = 35, 64%): seventeen patients (31%) completed both. Results from the analysis of T1/T2 well-being questionnaires are displayed in Table 1. 

For the well-being questionnaires, Wilcoxon Signed–Ranks tests were used due to the ordinal, non-parametric nature of the data. All assumptions were met. For the patient group, results show a statistically significant increase from before workshops (T1) to after (T2), in response to Questions 1, 2, 3 and 5 (Table 1). Self-reported ratings for Q1: "I enjoy taking time for myself" increased significantly post-workshops (*z* = 2.195, *p* = 0.028) with a small effect size (r = 0.28), representing a 40% positive change in ratings from T1/T2. There were also significant increases in outcome for Q2: “Taking time out for myself feels good” (*z* = 2.517, *p* = 0.012) between T1 and T2. The analysis suggested a medium effect size (r = 0.32) and 37% positive change between T1/T2 ratings. Q3: “I feel that I deserve to take time out for myself” increased significantly between T1 and T2 measurements (*z* = 2.419, *p* = 0.016) with a medium effect size (r = 0.31) and 47% positive change in T1/T2 ratings. A significant increase was also found for Q5: “Today I am in a good mood” between T1 and T2 (*z* = 3.839, *p* < 0.001), where there were 31 responses, with a medium effect size (r = 0.49) and a 63% positive change in T1/T2 scores. 

Thirty-five patients completed feedback forms for quantitative analysis. Results for Q1: “How much did you enjoy the event?” showed that 20 (57.1%) of the patients rated it either 5/5 or 4/5, with 5 the mode. In Q2: “How useful was the workshop?”, 14 (40.0%) patients rated it 3, which was the mode, 17 (48.6%) rated it 4 or 5/5. Q3: “How relevant was the workshop content to your own self-care practices?” was answered by 34 patients and the mode was 5 (*n* = 14), with 20 (58.8%) patient-participants rating it 4 or 5/5.

#### 3.2.3. The MDT

In total, thirty-four MDT participants attended the workshops and consented to the study. Twenty-seven (79%) completed both T1 and T2 well-being questionnaires (Table 2) and twenty-one (62%) staff completed the section of the feedback form used for quantitative analysis.

The Wilcoxon Signed-Ranks tests (Table 2) of the well-being questionnaires indicated that self-reported ratings for Q3: “I feel that I deserve to take time for myself” were significantly higher following the workshops (*z* = 2.814, *p* = 0.005), with a medium effect size (r = 0.38). There was a 41% positive change in T1/T2 ratings. An analysis found statistical significance in the increase of self-reported Q5: “Today I am in a good mood” (*z* = 3.043, *p* = 0.002), also with a medium effect size (r = 0.41). This was a 67% positive change in T1/T2 ratings.

Twenty-one MDT participants (62%) completed the section of the feedback form analysed quantitatively. Q1 “How much did you enjoy the event?” showed that 16 (76.2%) of the MDT rated it either 5/5 or 4/5 with 5 the mode. For Q2: ‘How useful was the workshop?’, 12 (57.1%) of the MDT rated it 4/5, making it the mode. 16 (76.2%) rated Q2 4 or 5/5. For Q3 “How relevant was the workshop content to your own self-care practices?” the mode was 4 (*n* = 9, 42.9%), with 15 (71.4%) of the MDT participants rating it 4 or 5/5.

### 3.3. Qualitative Results

Forty-three patients (78%) provided qualitative feedback, either in the form of written responses to open-ended questions on the feedback forms (*n* = 34, 62%) or by attending the focus group (*n* = 8, 2%). Thirty (86%) of the consenting MDT also provided qualitative feedback, with 21 (60%) completing the open-ended feedback questions and nine (26%) partaking in individual interviews. A summary of the qualitative data is provided below under headings of key themes (examples of quotes for each theme highlighted in Table 3, for extended table see Appendix A). Qualitative data collected via interviews and the focus group was regarding all runs of the workshops attended.

#### 3.3.1. Enjoyment

All of the patients found the workshops fun and enjoyable, however, some patients thought they had not yet reached their full potential and could improve.

From the MDT interviews, it was clear that attendees had a positive workshop experience and reported feeling happier. Furthermore, almost all of the MDT interviewed reported feeling more productive post-workshop.

#### 3.3.2. Recovery & Well-being

Patients consistently reported that they believed these workshops had the potential to address aspects of their recovery not currently in their care plans and that there was a definite need for them.

The MDT also thought that the workshops had the potential to support a more holistic recovery for patients and that the MDT modelling the importance of well-being was also very beneficial. Workshops addressed the importance of well-being and taking time to look after yourself for the MDT, highlighting their difficulty prioritising this, and how it is a major cause of burnout.

#### 3.3.3. Relationships

Workshops encouraged patients’ social activity, simultaneously decreasing isolation. Patients noted particular benefits of joining with staff including breaking barriers and bridging the social divide. However, this did not work as well as hoped in all workshops. Patients suggested this was due to facilitation and inter-activeness levels: When the workshop was more interactive with joint activities, it worked better.

The MDT remarked on how the ward environment can lead to difficult dynamics. They suggested the workshops helped foster more positive rapport between the MDT and patients. However, depending on the workshop, they also felt that they were there primarily for the patients.

#### 3.3.4. Content

Patients reported enjoying the emotions/communication and the yoga workshops. The make-up workshop was received extremely favourably by some patients, while others believed it had the potential to cause difficulties. Also, two different runs of the same workshop were perceived differently. Other comments related to not being sure what to expect beforehand, and questioning the physical aspects of some workshops as potentially unsuited to their stage of recovery. 

The MDT particularly enjoyed the yoga workshop, which used shared poses to engage both groups together in a peaceful way. Other workshops were thought to be too structured and instructor-led by some staff. Overall workshops where people were engaging and “doing” were received better.

#### 3.3.5. Structure

Patients suggested having a regular slot for well-being, alongside less-regular, larger workshops. This, they suggested, would encourage participation and decrease anxiety and uncertainty around attendance.

Staff also thought a higher frequency would be more beneficial. However, it was reported to be often difficult to take time out of their schedules to attend.

#### 3.3.6. Future Ideas and Improvements

Patients were interested in the idea of learning new practical skills and potentially learning skills from other patients and the MDT on the ward. For beauty-related workshops, participants preferred the idea of enhancing natural beauty. The focus group were more interested in supporting patients in learning basic day-to-day skills. All patients were very keen to have more facilitators who had also been through similar ED journeys. They also thought it was more beneficial when shared activities were conducted. 

The MDT heavily emphasized the importance of fun and playfulness in future workshops. They also thought shared activities should be the focus of the workshops. Furthermore, the MDT thought that explaining the details of the workshops beforehand was important for patients.

## 4. Discussion

The aim of the pilot study was to evaluate the experience of joint well-being workshops on ED patients and the clinical team providing specialist ED clinical services. Principle workshop goals were supporting patient-recovery and addressing staff work-place stress. Significant statistical results derived from patients’ self-report questionnaires pre and post-workshop showed enhanced mood and three areas of increased positive feelings related to taking time out for self namely; deserving, feeling good about, and enjoying. For the MDT, feelings of deserving taking time out for self, and overall mood post-workshop reached statistical significance. Feedback forms indicated patients evaluated workshops as enjoyable and relevant to self-care practices, and fairly useful, with potential improvements. MDT feedback found workshops enjoyable, useful and relevant. Six core themes were identified through qualitative thematic analysis from the patients’ focus group and MDT interviews: Enjoyment, well-being and recovery, relationships, content, structure and future ideas. Both groups highlight the potential benefits of joint well-being workshops, with some limitations being identified in the current implementation. Workshops addressed areas of well-being underpinning recovery and self-determination in patients and burn-out in staff, including; self-compassion, self-care feelings of deserving, social tension and isolation, mood, and motivation. 

### 4.1. Quantitative Analysis

The greatest effect size was found in mood improvement from analysis of pre- and post-workshop data, with a medium effect size for both patients (r = 0.49) and the MDT (r = 0.41). The immediate benefit of enhanced mood was echoed in feedback form ratings and qualitative themes. Specifically for patients, enjoyment focused on external, pleasurable or interesting activities. For the MDT, it was stress-free time, enjoying increased productivity and mood post workshops. Current research into well-being highlights that self-compassion and taking time for yourself relates to higher levels of happiness, alongside the reduction of negative mood and thinking [8]. What is new to the literature, to the best of our knowledge, is the finding in this study of an immediate positive impact on mood in both groups. This increases the potential usefulness of future workshops, perhaps targeting stressful times such as in-take or post-meal. 

Patient’s responses to Q1: “I enjoy taking time out for myself”, showed a small effect size (r = 0.28). As expected, pre-intervention ratings scores were low, consistent with evidence that patients fear self-compassion or treating themselves with kindness [7]. This key component of well-being is a particular patient difficulty [7], therefore any statistically significant change could have important implications in fear reduction around self-compassion. Implications are that regular workshops, or longer duration of this type of treatment, could potentially have a greater impact. This score did not significantly change for staff, consistent with the literature of non-ED controls not having a fear of self-compassion [7].

A medium effect size (r = 0.32) was found for patients for Q2: "Taking time out for myself feels good" suggesting the workshop produced a shift in patient self-compassion. This is important in the complex ED patient profile where shame and guilt feature strongly, and self-compassion is lacking [27], particularly in adolescence, where self-compassion has been found to be positively correlated with body satisfaction [28]. This suggests workshops could influence body perception. Interestingly, MDT ratings recorded no change for Q2, indicating no fear of self-compassion and prior recognition that taking time out for self feels good (M = 4.56).

Both staff and patients reported a lack of deserving associated with self-care, in line with the literature [27,29]. It was of note that the results suggest this improved post-workshop, with increased ratings in each group of Q3: “I feel that I deserve to take time out for myself”. Medium effect sizes were observed (patients: r = 0.31; MDT: r = 0.38). Workshops seemed to promote a shift in attitudes to self-care. Potentially, gradual and frequent exposure through workshops will increase patient self-compassion and well-being in both patients and staff.

No significance was found in either group for Q4: “I don’t take time out for myself very often” (Table 1 and Table 2). This is a likely fact of hospital life, and we would therefore not expect any change.

### 4.2. Qualitative Analysis

Thematic analysis of qualitative data from both groups identified six core themes: Enjoyment, recovery and well-being, relationships, content, structure and future ideas (Table 3). The theme of enjoyment re-enforced statistical analysis, with enhanced well-being experienced individually and in the group dyad, with a sense of shared enjoyment. Mood benefits linked to well-being and self-compassion are important, as discussed previously.

Recovery and well-being, the second theme, was felt to be universally applicable, a message the workshops successfully promoted. Recovered ED patients have identified well-being as a fundamental aspect of their recovery [4], and the importance of encouraging and promoting well-being in ED treatment was a major theme for patients and the MDT. There was a recognition that everyone can enhance psychological functioning irrespective of illness. However, while staff described the benefit from a stress-free “oasis”, they also focused on patient well-being during interviews, and felt their primary function during workshops was still to support patients. It would be important to explore with the MDT how this perception could be changed, as staff felt the ability to take time for themselves was directly correlated to burnout.

Relationship dynamics, the third theme, featured frequently with both groups and workshops provided an opportunity to build empathy and trust. Well-being was felt to have a huge potential for recovery. Research suggests reducing patient isolation is also important in recovery [11], and all participants noted this benefit. The MDT discussed a strained environment was previously identified in EDUs [12], and workshops were seen as breaking down barriers and improving communication and relationships. Reducing relationship tension also reduces workplace stress in EDUs [12]. However, the analysis showed relationship improvements differed due to facilitators and levels of interactive activities in workshops, which is useful knowledge for future workshop design. 

The fourth theme, content, was felt to be appropriate by all participants. It was mainly well-received and contributed to well-being: Yoga was the favourite workshop. However, the same content by the same facilitator (a professional make-up artist) was perceived differently on different days, and staff linked this to the dramatic variation in daily atmosphere dealing with a life-threatening illness. Patients felt that learning skills to take away was important, whilst acknowledging this may not appeal to staff: Engaging both groups equally in content is challenging. 

The fifth theme of structure reflected anxiety associated with the unpredictability of the workshop programmes, but only for patients. Whilst challenging participants to push boundaries is positive in promoting coping and resilience, the sporadic scheduling of workshops was potentially unhelpful in a practical sense. Patients requested regular workshops and regular smaller groups for the time between workshops. The MDT expressed difficulty finding time to attend. To reduce patient anxiety and increase the potential for all participants to attend, a regular schedule should be implemented enabling all participants to block time in diaries and treatment plans.

Both groups enthusiastically discussed ideas for future workshops, which constituted the final theme. Ideas focused on more creative, interactive, and less directed workshops. In particular, patients found recovered facilitators with an ED very helpful role models, while staff emphasised the importance of fun. The high level of engagement suggests an opportunity to involve participants and enhance proactiveness, motivation and self-agency.

### 4.3. Future Research and Limitations 

Future research could examine the effect of collaboration further by comparing staff only, and patients only workshops, investigating the issue of staff feeling that the space was primarily for patients. Furthering our understanding of self-compassion is indicated, possibly through the use of the Self-Compassion Scale [30] before and after well-being workshops. Furthermore, the interpretation of the reported improved relationships reducing stress in staff, and patients, should be investigated. A longitudinal study linking well-being interventions and recovery to BMI would also be interesting. With the frequency of patients with high autistic traits, it might be interesting to research how the workshops were received differently.

One limitation of this study is that a well-being questionnaire was only created after the benefits to both groups were clear, limiting the valid sample size. Furthermore, due to the naturalistic nature of this study, there were varying relationships between each member of the groups and this aspect was not controlled for. Another limitation was the wording of Q4 on the well-being questionnaire where the double negative confused some participants. Furthermore, this study only captures short-term effects. The concept of well-being may have varied over our wide age-range; however, for this pilot study, we wanted to be inclusive of all age ranges and aimed to choose workshops that were relevant to all.

## 5. Conclusions

Overall, the results from our study provide evidence that regular implementation of well-being workshops would provide well-being benefits for both patients and staff. For patients, our results support increased mood, a reduction in fear towards self-compassion, improved relationships and decreased isolation. For the clinical team, results show enhanced mood, increased productivity and improved relationships. However, in targeting MDT burnout specifically, we need to ensure staff attend for themselves and not to support patients. Improvements in workshop content, timing and facilitation are also indicated, with patients keen to be actively involved in implementing improvements. Evidence of a high level of engagement was encouraging as motivation and self-agency underpin aspects of well-being and recovery. 

To our knowledge, this pilot study is the first to evaluate the effects of implementing joint well-being workshops for patients and staff in an EDU and its short and potential long-term benefits. Furthermore, it is the first to analyse varied well-being workshops in any mental-health specialist field, and to use a mixed-method analysis. Bringing the clinical team and patients together for well-being workshops in any clinical setting is under-utilised and under-researched, and we believe this mixed-methods pilot study strongly supports a role for regular well-being workshops and further research.

## Figures and Tables

**Table 1 brainsci-09-00247-t001:** A table showing patients outcome measures for T1 and T2, *z*-scores and *p*-values.

Outcome Measures	Pre Workshop (T1)			Post Workshop (T2)			*z*	*p*
	*n*	Mean	*SD*	*n*	Mean	*SD*		
Q1 I enjoy taking time for myself	34	3.00	1.15	30	3.42	1.31	2.195 *	0.028
Q2 Taking time out for myself feels good	34	2.91	1.16	30	3.29	1.27	2.517 *	0.012
Q3 I feel that I deserve to take time out for myself	34	2.06	1.10	30	2.67	1.38	2.419 *	0.016
Q4 I don’t take time out for myself very often	34	3.24	1.39	30	3.12	1.25	–0.554	0.580
Q5 Today I am in a good mood	35	2.69	1.02	31	3.48	1.12	3.839 **	0.000

* Significance at 0.05; ** Significance at <0.001.

**Table 2 brainsci-09-00247-t002:** A table showing the MDT’s outcome measures for T1 and T2, *z*-scores and *p*-values.

Outcome Measures	Pre Workshop (T1)			Post Workshop (T2)			*z*	*p*
	*n*	Mean	*SD*	*n*	Mean	*SD*		
Q1 I enjoy taking time for myself	34	4.50	0.71	27	4.56	0.75	0.816	0.414
Q2 Taking time out for myself feels good	34	4.53	0.61	27	4.56	0.58	0.378	0.705
Q3 I feel that I deserve to take time out for myself	34	4.00	0.89	27	4.44	0.64	2.814 *	0.005
Q4 I don’t take time out for myself very often	34	3.35	1.07	27	3.56	1.09	1.567	0.117
Q5 Today I am in a good mood	34	3.41	0.89	27	4.00	1.11	3.043 **	0.002

* Significance at 0.05; ** significance at <0.01.

**Table 3 brainsci-09-00247-t003:** A table of qualitative examples for each theme for the MDT and patient groups.

Theme	Patient Examples	MDT Examples
Enjoyment	“I found it a lot of fun and a great positive distraction”	“I actually really enjoyed the workshops. Taking that time out and doing something and coming back to work, it is so much more productive”
Recovery and Well-being	“We are kind of reinstating our true identify away from this illness and I think that… in every workshop, in any situation, you do at the end of it take something back and reflect upon it and that impacts in the longer term beyond anorexia”	“I really feel that burn-out is directly correlated with the ability for staff to take time out or for staff to look after themselves in their well-being”
Relationships	“I guess if they are interacting with people, that’s the big one, and it’s hard because its sort of instinctive, you want to isolate a lot of the time”	“The relationships and dynamics can be quite difficult between the staff and the patients….it builds up therapeutic relationship quicker”
Content	“Focusing on something pleasurable for no reason other than its something that makes you happy”	“A simple exercise like standing on one leg on your own and realising you are really wobbly and then you stand on one leg and everyone supports each other with a single hand and that is all you need and suddenly you realise we can all stand really steady in a circle when we support each other”
Scheduling	“You could have your little things going on all the time and it gets everyone engaged and feeling secure, it gets the staff and patients used to doing stuff together…”	“Having more. I think having more, having them frequently as well”
Future Ideas	“I would rather have more sort of activity led ones, sort of engaging ones…something that can inspire and you can take outside as well.”	“If you can do something like creative and just fun and getting like messy and maybe even making something”

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
