# Peer review of "Well-Being Workshops in Eating Disorder Wards and Their Perceived Benefits to Patients and the Multi-Disciplinary Team: A Pilot Study"

_brainsci, 2019, doi:10.3390/brainsci9100247_

Round 1

Reviewer 1 Report

Manuscript number: brainsci 597440

Manuscript title:  Well-Being Workshops on Eating Disorder Wards and their perceived

benefits to patients and the Multi-disciplinary Team: A Pilot Study

This research focus a very important topic In the context of EDs: Staff burn-out linked to difficult relationships with patients, and poor self-care of these patients concerning all types of treatments. The research is well done from a methodological point of view and paper is well written in all its section. I only suggest some minor revisions before publishing the paper.

Abstract

You use for the first time the abbreviation MDT (page 1, line 14) but you don’t specify before what it refers to.

Methods

Concerning participants you wrote (page 2, lines 77-78): “There was no exclusion criteria for the MDT”.  

I believe that there are some clarifications to do. For example, you have to specify what type of relationship exist between the MDT and the selected patients: Are all the patients new patients for the MDT? Alternatively, are these patients the same patients with which  the team worked for many years? This aspect could become a bias with respect to the aims of your research.

What about patients? I think that there are some inclusion or exclusion criteria to consider: What about age of illness? How many treatments these patients underwent? Are there any differences concerning the concept of well-being between patients with 18 or 60 years old? I suggest to add some specifications concerning this point.

Author Response

We are grateful to reviewer for positive comments and great suggestions. We have tried our best to address all comments reviewer raised.

Abstract

You use for the first time the abbreviation MDT (page 1, line 14) but you don’t specify before what it refers to.

Thank you this has been spelled out “Multi-disciplinary team” (MDT).

Methods

Concerning participants you wrote (page 2, lines 77-78): “There was no exclusion criteria for the MDT”.  

I believe that there are some clarifications to do. For example, you have to specify what type of relationship exist between the MDT and the selected patients: Are all the patients new patients for the MDT? Alternatively, are these patients the same patients with which  the team worked for many years? This aspect could become a bias with respect to the aims of your research.

We thank the reviewer for the comment. Because we have national eating disorder department we have a very mixed patient group from workshop to workshop. For example, national patients who come from different parts of the country and local patients who might have repeated admissions, therefore it is always a mixed group in terms of experiences and potential relationships between each group. Some will have been new MDT as well as patients, some would have been well-known to the service.

We agree that not collecting this is a limitation, we have highlighted it in the methods and in the discussion. 

What about patients? I think that there are some inclusion or exclusion criteria to consider: What about age of illness? How many treatments these patients underwent? Are there any differences concerning the concept of well-being between patients with 18 or 60 years old? I suggest to add some specifications concerning this point.

 All questions the reviewer raised are highly relevant for our future large-scaled studies. For this pilot, as we have highlighted above we had mixed group. This has now been added to the limitations.

Reviewer 2 Report

Overall, I enjoyed reviewing this manuscript and believe it could make a notable contribution to the literature of the major theme of well-being in anorexia nervosa (AN). The paper evaluates the impact of joint well-being workshops on patients with AN and multi-disciplinary team (MDT) members with mixed quantitative and qualitative approaches. Data of 55 adults with AN and 34  MDT members were studied. The main limitations of this study are the small number of participants, the absence of a control group of patients for the quantitative evaluation, and the absence of separate groups of patients and MDT members to show the effects of the joint group. However, the authors entitled the paper "a pilot study" and acknowledged and discussed these limitations. I propose a minor revision according to some of these suggestions:

Materials and Methods section:

Could you explain in more detail

-How and why you chose the themes of the well-being groups? Could you particularly specify the link between these themes and the well-being?

-Why did you choose 4 different themes and not a single theme?

-How did the participants choose their groups?

It is not clear whether the authors decided on the number and the theme of the workshops or if the participants chose a theme among those proposed. Besides their choice determined the number of workshops for each theme. In the second case the number of workshops (8) and their themes should be placed in the results section.

Results section:

P.3 lines 131-132, in the Demographics results, the authors wrote that "...scored highly on the Autisme Spectrum Quotien (AQ10) at admission" but this questionnaire was not mentioned in the 2.3 Procedures and Measures section.

There are 2 typing errors in the Table 1. Firstly, the significance of p value for the Q2 was not ** but only * (P=0.012). Secondly, ** corresponds to a significance at <0.01 and not <0.001.

P.5 line 191, it is an additional "of" in the sentence "a summary of of the qualitative data..."

In the Content section of the Qualitative Results, the authors wrote P.7 lines 221-222 "...some patients,while others...". It would be interesting to better characterize these different types of patients (with BMI, duration of the disease, ASQ10 or other audit measures) and more generally to better characterize the profil of patients in each workshops' theme. Moreover there was no explanation here or in the discussion section about the fact that nobody chose the colour workshop.

Discussion section:

P.7 line 257, the sentence "All results highlight the potentiel benefits of joint well-being workshop" must be nuanced. Only some elements exist in favor of the joint workshops and this point will have to be explored further in the future.

Once again, I appreciate your efforts in doing this study and wish you the very best.

Author Response

The main limitations of this study are the small number of participants, the absence of a control group of patients for the quantitative evaluation, and the absence of separate groups of patients and MDT members to show the effects of the joint group. However, the authors entitled the paper "a pilot study" and acknowledged and discussed these limitations. I propose a minor revision according to some of these suggestions:

We completely agree with the fair comment the reviewer raised here. We wanted to check the feasibility of this intervention to scale it up for bigger study with reasonable pilot data.

Materials and Methods section:

Could you explain in more detail

-How and why you chose the themes of the well-being groups? Could you particularly specify the link between these themes and the well-being?

We were largely informed by the excellent work on well-being which has been ongoing in cancer patients for many years. Principal investigator (KT) is a patient with lived experience of cancer and has established strong links with cancer health psychology colleagues. She has tried to adjust this knowledge of well-being and apply it to an eating disorder context.  Similarly to the work being done with cancer patients, we wanted to shift the focus from illness specific themes to generic well-being topics, which might be very relevant for eating disorder patients as well.

-Why did you choose 4 different themes and not a single theme?

Because of the pilot nature of the study, we wanted to explore different themes and evaluate the patient and staff satisfaction, relevance and importance of each to make informed decisions with the future work.

-How did the participants choose their groups?

Participants were informed of all upcoming workshops in advance by posters throughout the ward and through verbal communication. Staff were additionally emailed reminders and everyone was welcome to join. This has been detailed in the methods section, thank you for highlighting it.

It is not clear whether the authors decided on the number and the theme of the workshops or if the participants chose a theme among those proposed. Besides their choice determined the number of workshops for each theme.

Due to the pilot nature of this study, funding was not available for external facilitators of the workshop. Therefore all facilitators were volunteers who were also professionals in the theme of their workshop. We wanted to utilise an external facilitator to ensure no bias in relationships. Workshop numbers and choices were determined by the availability of the volunteers. We hope this has now been made clearer in the edits to the manuscript.

In the second case the number of workshops (8) and their themes should be placed in the results section.

This has been addressed in both the quantitative results section and the qualitative results section.

Results section:

P.3 lines 131-132, in the Demographics results, the authors wrote that "...scored highly on the Autism Spectrum Quotien (AQ10) at admission" but this questionnaire was not mentioned in the 2.3 Procedures and Measures section.

This has now been addressed. 

There are 2 typing errors in the Table 1. Firstly, the significance of p value for the Q2 was not ** but only * (P=0.012). Secondly, ** corresponds to a significance at <0.01 and not <0.001.P.5 line 191, it is an additional "of" in the sentence "a summary of of the qualitative data..."

Thank you, these mistakes have now been edited.

In the Content section of the Qualitative Results, the authors wrote P.7 lines 221-222 "...some patients, while others...". It would be interesting to better characterize these different types of patients (with BMI, duration of the disease, ASQ10 or other audit measures) and more generally to better characterize the profile of patients in each workshops' theme.

This is an excellent idea and we will take it in to account for future study design.

 Moreover, there was no explanation here or in the discussion section about the fact that nobody chose the colour workshop.

It was mentioned in the methods section that we started to collect quantitative data shortly after the introduction of the workshops. Unfortunately, the colour workshop was one of the first and therefore there was no quantitative data collected. However, the qualitative data capture this as many of the staff and a few of the patients who have since been re-admitted were able to report on it.

Discussion section:

P.7 line 257, the sentence "All results highlight the potential benefits of joint well-being workshop" must be nuanced. Only some elements exist in favour of the joint workshops and this point will have to be explored further in the future.

This has been edited and made clearer.

Once again, I appreciate your efforts in doing this study and wish you the very best.